# Understanding Adaptive Skills in Borderline Intellectual Functioning: A Systematic Review

**DOI:** 10.3390/ejihpe15030040

**Published:** 2025-03-20

**Authors:** Cristina Orío-Aparicio, Cristina Bel-Fenellós, Carmen López-Escribano

**Affiliations:** Department of Research and Psychology in Education, Faculty of Education, Complutense University of Madrid, 28040 Madrid, Spain

**Keywords:** adaptive functioning, adaptive behavior, borderline intellectual functioning, borderline intellectual disability, borderline intelligence, systematic review

## Abstract

Borderline intellectual functioning (BIF) is characterized by an IQ typically ranging from 70 to 85, combined with deficits in adaptive functioning. Despite its prevalence, individuals with BIF are often excluded from diagnostic and support systems, which traditionally focus on strictly defined intellectual disabilities. This article presents a systematic review conducted across the ProQuest, WoS, SCOPUS, and EBSCOhost databases, aiming to develop a profile of the adaptive functioning in individuals with BIF. A total of 64 documents published from 2012 to the present were included, all of them addressing BIF and adaptive functioning skills, and quality was assessed using the SSAHS tool. The findings presented are synthesized according to conceptual, social, and practical domains and reveal that individuals with BIF experience widespread difficulties across the conceptual, social, and practical domains. Additionally, societal barriers, primarily limiting access to support services, persist. However, there are emerging resources aimed at supporting this population, such as legislative efforts to facilitate their integration into the labor market. The implications and limitations of the findings are discussed, highlighting the need to consider the adaptive functioning skills of individuals with BIF.

## 1. Introduction

Borderline intellectual functioning (BIF) can be defined as a “health meta-condition that requires specific socio-health, educational, and legal attention, characterized by diverse cognitive dysfunctions associated with an Intelligence Quotient (IQ) between 71 and 85, which determine a deficit in the person’s functioning concerning activity restrictions and limitations in social participation” ([70]).

However, neither the diagnostic manual published by the [9] ([9], DSM-5-TR) nor the World Health Organization classification of diseases and related health problems (ICD-11, [98]) include BIF as a specific diagnostic category or among the severity levels for intellectual developmental disorder. Nevertheless, they consider it under “Additional Conditions or Problems That May Be a Focus of Clinical Attention” or “Other Specified Symptoms and Signs Relating to Cognition,” respectively. Likewise, the manual of the American Association on Intellectual and Developmental Disabilities (12th ed., Schalock et al., 2021) defines BIF as “a group of individuals who do not technically meet the criteria for a diagnosis of ID but share many characteristics and support needs of those who do”.

None of these manuals provide information on the specific characteristics of this condition or guidelines for its evaluation. However, they do emphasize the importance of a rigorous assessment of intellectual and adaptive functioning to distinguish BIF from mild intellectual disability.

A recent scoping review ([56]) aimed to map the available evidence on BIF, focusing on its scope and volume, identified a total of 138 documents addressing BIF, including academic articles and gray literature. The 138 documents included in the scoping review were categorized according to the diagnostic criteria proposed by the DSM-5-TR and the AAIDD: intellectual functioning, adaptive functioning, additions for comprehensive evaluation, and more general articles. In this article, we will delve into the documents focused on adaptive functioning.

The concept of adaptive functioning (or adaptive behavior) as we understand it today emerged in the mid-20th century, when [32] ([32]) defined it through a three-factor structure comprising social, practical, and conceptual domains. Later, in 1992, this concept appeared for the first time as part of the definition of intellectual disability in the ninth edition of the manual of the American Association on Mental Retardation ([4]), now known as the American Association on Intellectual and Developmental Disabilities (AAIDD) ([44]). This manual defined adaptive deficits across ten areas: Communication, Self-Care, Home Living, Social Skills, Community Use, Self-Direction, Health and Safety, Functional Academics, Leisure, and Work. Two years later, the [7] ([7]) modified the diagnostic criteria for intellectual disability (ID) to incorporate adaptive functioning in addition to intellectual functioning, requiring deficits in at least two of the ten areas proposed by the aforementioned AAMR manual.

Later, the 10th edition of the manual published by the American Association on Mental Retardation ([5]) reintroduced Heber’s structure. However, it was not until 2013, with the DSM-5, that the [8] ([8]) modified its definition of adaptive functioning to this three-domain model, a definition that remains current in the DSM-5-TR. On the other hand, the [6] ([6]) speaks of adaptive behavior and defines it as “the collection of conceptual, social, and practical skills that are learned and performed by people in their everyday lives”.

These two manuals, currently in force, align in their definition of the three main domains of adaptive functioning. The conceptual (academic) domain encompasses competencies related to memory, language and literacy, mathematical reasoning and numerical concepts, acquisition of practical knowledge, problem-solving, judgment in novel situations, self-direction, and time management. The social domain includes awareness of others’ thoughts, feelings, and experiences; empathy; interpersonal skills; friendship abilities; social judgment and responsiveness; and the ability to follow rules, obey laws, and avoid victimization, among others. Finally, the practical domain comprises aspects related to daily living activities, such as personal and healthcare activities, occupational skills, money management, recreation, self-management of behavior, task organization, schedules or routines, and transportation.

Several scales dedicated to assessing adaptive functioning are in use today, such as, for example, the Vineland Adaptive Behavior Scales, Third Edition (VABS-3, [77]), the Diagnostic Adaptive Behavior Scale (DABS, [85]), or the Adaptive Behavior Assessment System, Third Edition (ABAS-3, [28]).

Evaluating adaptive functioning is crucial in the diagnosis of borderline intellectual functioning (BIF). First, diagnostic manuals emphasize the importance of a thorough assessment of both intellectual and cognitive functioning to differentiate BIF from mild intellectual disability (ID). Second, although adaptive functioning and IQ are related constructs, they are independent of each other ([84]; [47]). In other words, not everyone with limitations in intellectual functioning necessarily has limitations in adaptive functioning, and vice versa ([82]). This makes evaluating both constructs essential for making a rigorous diagnosis, especially in the case of BIF, as it has been observed that the correlation between adaptive functioning and intellectual functioning decreases as IQ increases ([3]). In fact, some authors suggest that, in some cases, individuals with an IQ just above 70 but with limitations in adaptive functioning may benefit from specialized services aimed at individuals with mild ID, if diagnosed as such by a professional based on their clinical judgment ([73]). Finally, beyond the diagnostic process, evaluating adaptive functioning is essential because it provides information on individuals’ daily skills and areas where support is needed ([83]).

It is important to note that adaptive functioning assesses an individual’s actual performance in daily activities. That is, it does not focus on the individual’s capacities or maximum potential but rather on the real skills demonstrated in everyday life ([82]). To this end, the scales used to evaluate adaptive functioning refer to observable and measurable behaviors.

However, individuals’ capacities and skills are, of course, essential for good adaptive functioning. These are the skills examined in this systematic review, as most of the included articles do not assess adaptive functioning per se but rather describe various characteristics of the BIF population that may be related to this construct. Therefore, this article focuses on the adaptive skills of individuals with BIF rather than on adaptive functioning itself.

Therefore, this review aims to address the following question: What are the patterns of adaptive skills in individuals with BIF across the conceptual, social, and practical domains? This focus is particularly relevant given the difficulty in accessing the population with borderline intellectual functioning.

## 2. Materials and Methods

### 2.1. Design

This systematic review builds on a scoping review previously conducted by the authors ([56]), which followed PRISMA-ScR ([87]) and the framework proposed by the Joanna Briggs Institute ([62]). The aim was to analyze the body of research on borderline intellectual functioning.

In that scoping review, documents were searched across various databases. In addition, gray literature was identified through sources such as Google and Google Scholar.

The query string used was “funcionamiento intelectual límite” OR “capacidad Intelectual límite” OR “CI límite” OR “cociente intelectual límite” OR “coeficiente intelectual límite” OR “retraso mental límite” OR “borderline Intellectual functioning” OR “subaverage intellectual functioning” OR “borderline developmental disability” OR “borderline intellectual disability” OR “borderline IQ” OR “borderline learning disability” OR “borderline mental retardation” OR “minor intellectual disability”.

Finally, that previous scoping review included a total of 138 documents on borderline intellectual functioning published between January 2012 and June 2024. Then, documents were categorized using the diagnostic criteria for intellectual disability proposed by major manuals ([9]; [74]), given the lack of clear, specific criteria for BIF: intellectual functioning, adaptive functioning, and additions for a comprehensive evaluation. In addition, a category of more general documents that approach BIF in a holistic manner was included.

A complete description of the methodology can be found in the cited article and its prior registration on the Open Science Framework (https://osf.io/km2rp accessed on 16 March 2025).

### 2.2. Research Question and Objective

The main research question formulated to guide this systematic review was as follows: What are the patterns of adaptive skills in individuals with BIF across the conceptual, social, and practical domains? Therefore, the main objective, based on the reviewed literature, is to build a profile of the adaptive skills of people with BIF.

### 2.3. Document Selection

As explained earlier, the studies included were extracted from a previous scoping review ([56]). To answer the research question of this article, only documents related to adaptive skills were analyzed. As explained by [56] ([56]), most of the documents do not specifically focus on adaptive functioning in individuals with BIF. Nevertheless, many of them address topics that fall within some of the domains of this concept, as explained in the Introduction.

Thus, the inclusion criteria for this systematic review are as follows:Documents published from 2012 onwards.Studies on borderline intellectual functioning as a specific group (even if they use other terms such as “slow learners” or “borderline intellectual disability”).No limits on age, gender, nationality, or other characteristics of the sample.Documents addressing adaptive skills.Documents published in English or Spanish.

The initial full-text examination was conducted by the first author. The second author reviewed the first half of the documents, while the third author reviewed the second half. Overall, an agreement rate of 87% was achieved. In cases of disagreement, all three authors collaboratively re-evaluated the text and resolved the issues through consensus.

A PRISMA flow diagram of the study selection process is shown in Figure 1.

### 2.4. Data Collection and Synthesis Process

The data collection process followed the same procedure as the document selection, with the first author performing the extraction and the other two authors corroborating. The author and year of each document were collected, along with their origin (i.e., whether they were scientific articles, book chapters, gray literature, etc.), their location, study design, the terms used to refer to BIF, and the sample employed (age and size).

Additionally, the research objectives of each article, as well as their main results, were collected. This information is synthesized in the Results section, presented according to the three domains of adaptive functioning (conceptual, social, and practical). To present the results in a logical and comprehensible manner, within each of these sections, the information are organized using bolded headers. These headers were determined by grouping articles that addressed similar recurring topics in the literature.

### 2.5. Quality Assessment

Given the methodological diversity within the documents included in this review, the quality of the studies were assessed using the Scale to Evaluate Scientific Articles in Social and Human Sciences (SSAHS, [43]). This is a scale composed of 19 items referring to eight dimensions: title and abstract (3 items), introduction (2 items), methodology (4 items), results (3 items), discussion (3 items), references (1 item), appendices (1 item), and style and format (2 items). Each of these items is evaluated based on the quality of the study using a 5-point Likert scale, where 1 = very low level, 2 = low level, 3 = medium level, 4 = medium-high level, and 5 = very high level. Thus, the minimum score that can be obtained is 19 points, and the maximum is 95. The reliability and construct validity of the scale are optimal, with reliability scores of **α**= 0.937 and construct validity scores of V > 0.80. This scale has been used in other similar systematic reviews, such as [52] ([52]), [65] ([65]), [71] ([71]), or [92] ([92]).

The first author evaluated all documents without using automation tools. Subsequently, the second author evaluated the first half of the documents and the third author evaluated the other half. The total mean score for each of the documents was calculated by including both scores.

## 3. Results

### 3.1. Main Characteristics

#### 3.1.1. Quality

The quality scores given to each study can be seen in Table 1, Table 2, Table 3 and Table 4. As can be seen, most of the studies scored between high and very high quality. In addition, most of the included studies are articles published in peer-reviewed journals or at scientific conferences, which ensures, to some extent, the quality of their content.

In the case of the gray literature, the quality of the included studies is optimal, although sometimes the scores are between 3 (medium level) and 4 (high level), as they correspond to abstracts of conference papers that do not contain all the information necessary for their evaluation.

#### 3.1.2. Location

Documents written by authors from Europe (n = 41), South America (n = 1), North America (n = 6), Asia (n = 14), and Oceania (n = 2) were identified. The specific countries can be seen in Table 1, Table 2, Table 3 and Table 4. If the article identified the country where the study was conducted, that country was documented. Otherwise, the country of the first author was recorded.

#### 3.1.3. Study Design

Of the 64 documents included in this review, the majority (n = 34) employed a quantitative study design. There were also five case studies, six mixed-methods papers, and five literature reviews, followed by three qualitative articles and eight articles with mainly theoretical content. Finally, this review also included two practical guides, one meta-analysis, and one legal document.

#### 3.1.4. Population and Sample

Although this article uses the term “borderline intellectual functioning” (or BIF) for ease of understanding and readability, as it is the most widely used terminology in the literature, not all authors apply this term. Among the documents included in this systematic review, 36 used the term “borderline intellectual functioning”, and other papers used other synonymous terms, such as “borderline intelligence” (n = 7), “borderline intellectual disabilities” (n = 4), “borderline intellectual ability” (n = 5), “borderline learning disabilities” (n = 1), “low intellectual functioning” (n = 1), “low and below average IQ” (n = 1), or “borderline IQ” (n = 1), all referring to an IQ between 70 or 71 and 84 or 85.

Additionally, other documents referred to this population as “slow learners” (n = 6), attributing, in this case, a broader IQ range. This range varies from more restrictive cases of 71 to 84, through ranges like 70–84 or 75–89, to the widest definitions, which include IQs from 70 to 90. Finally, two of the articles used the “slow learners” and “BIF” terms interchangeably within their texts.

Some of the documents used population-based samples (n = 48), while others conducted studies based on the existing literature (n = 16). Among those that used population samples, fifteen focused on children (ages 3 to 11), six on adolescents (ages 11 to 20), eight on both children and adolescents, two examined aspects related to young adults (ages 20 to 40), nine included adults, and eight studies included participants of all ages.

Looking again at the papers that used population-based samples, 18 focused exclusively on a sample of individuals with BIF (investigating them directly and/or their families). Other documents (n = 14) examined a general sample in which individuals with BIF were subsequently identified, nine included samples composed of individuals with both BIF and intellectual disabilities (mainly mild intellectual disabilities), and five compared individuals with BIF to those with typical development. Additionally, two articles included samples composed of other stakeholders, interviewing professionals and company managers about individuals with BIF.

### 3.2. Objectives and Main Findings

As explained in [56] ([56]), some papers addressed several topics included in different categories of the BIF profile (intellectual functioning, adaptive functioning, comorbidities…). However, our analysis will be limited to the portions of the documents pertaining to adaptive skills.

Consequently, each domain of adaptive functioning is presented below, along with an analysis of the objectives and main findings of the articles included in each category. The documents have been included based on their primary themes, considering the definitions provided by the main diagnostic manuals for each domain of adaptive functioning.

#### 3.2.1. Conceptual Domain

According to the manuals published by the [9] ([9]) and the [6] ([6]), the conceptual domain encompasses aspects related to memory, language, literacy, mathematical reasoning, problem-solving, concepts of time and money, and self-direction.

We identified 15 documents within this domain, which can be divided into two categories: one focusing on the academic profiles of individuals with BIF and the other on interventions aimed at improving conceptual skills in this population. The first category helps establish an academic profile, while the second presents practical interventions and their results.

**Academic profiles:** The documents included in the academic profile subgroup served various objectives. Some of them aimed to identify the relationship between academic performance and cognitive profiles. For example, [1] ([1]) conducted their study on university students, finding a weak positive correlation between IQ and average academic outcomes and concluding that performance in university is attributable to cultural factors rather than the presence of BIF. [86] ([86]), on the other hand, focused on children. In their study, BIF children scored lower than the chronological age-matched comparison group in both academic and cognitive tasks. They also examined the developmental pace of the BIF group in both arithmetic and cognitive tasks, showing that the BIF group was slower in arithmetic tasks and word reading. However, both groups progressed at a similar rate in cognitive tasks.

Other articles delved further into specific skills. For instance, [16] ([16]) and [42] ([42]) focused on literacy skills. The former sought to examine reading skills (reading fluency, accuracy, and comprehension parameters) in children with either BIF or mild ID and found that BIF children performed worse on reading tasks compared to typically developing children but better than children with mild ID. Meanwhile, [42] ([42]) stated that children with BIF exhibit difficulties in skills related to metaphorical reasoning comprehension. On the other hand, [79] ([79]) explored mathematical skills, suggesting that they were consistent with the IQ scores and were significantly lower in students with BIF than in their typically developing peers. The greatest challenges were identified in the areas of operation tasks and working memory.

[36] ([36]) found through a review of national and international studies that BIF is characterized by difficulties in short-term, working, and long-term memory. Students with BIF exhibit challenges with concentration, which makes it difficult for them to learn abstract content, as well as deficits in executive functions. These deficits lead to low self-awareness and, as a result, students with BIF tend to adopt learning strategies centered on rote memory, meaning the literal memorization of content without a deep understanding of it, which causes their learning to remain only sustained in the context in which it is learnt, limiting its transferability to other situations and leading to rapid forgetting. Furthermore, [20] ([20]) highlight that students with BIF experience significant difficulties with writing and mathematics, along with generally poor performance across all subjects, which is often accompanied by grade retention.

Finally, [53] ([53]) proposed a non-psychometric approach for identifying BIF in rural schools, facilitating these processes in contexts where access to IQ tests is limited. Their approach involves conducting teacher interviews, reviewing grades, observing the learning process, and administering an ad hoc test.

In summary, the academic profile of individuals with BIF appears to be characterized by various academic difficulties in literacy, mathematics, and other cognitive aspects such as memory and attention. These challenges lead to low academic performance and feelings of frustration, low self-esteem, or school dropouts within this population.

**Interventions:** Among the articles reporting on interventions aimed at improving aspects of the conceptual domain of adaptive functioning, various focal points were identified.

On the one hand, [2] ([2]) and [17] ([17]) designed interventions using computer programs. The first ones aimed to demonstrate the effectiveness of developing mathematical thinking through computer programs based on learner control and program control strategies. Their study demonstrates that both computer programs equally improved mathematical thinking compared to the control group. The latter, in turn, focused on reading skills, using a computer reading program based on the dual-route model ([17]). It was demonstrated that their computer program improved text reading accuracy, as well as word and non-word reading. However, there were no significant improvements in word and non-word reading speed.

Similarly, [41] ([41]) show how behavioral modification, specifically through positive reinforcement, can increase the percentage of compliance and task completion in children with BIF. More broadly, [45] ([45]) demonstrated the effectiveness of a comprehensive academic intervention in developing various developmental skills (such as adaptive, communication and cognitive skills). However, the intervention was not significant in terms of personal–social and motor skills.

Additionally, [76] ([76]) explored the benefits of music therapy on the academic functioning of individuals with BIF, particularly in the areas of reading and writing as well as mathematical skills.

Finally, two articles propose designs to improve aspects related to the conceptual domain of adaptive functioning in populations with BIF but do not implement them, making their contribution primarily theoretical.

One of them ([29]) proposes a framework for incorporating tablet technology into the teaching/learning process for slow learners. On the other hand, [67] ([67]) present the design of the NUMERATica application, which employs gamification to teach numerical concepts.

Overall, the documents indicate that different methodologies, such as computer programs or music therapy, can enhance skills in the conceptual domain of students with BIF, including mathematical thinking and literacy skills. Furthermore, the incorporation of other innovative methodologies that integrate technology and gamification is suggested.

**Table 1 ejihpe-15-00040-t001:** Main characteristics of documents addressing the conceptual domain.

Author (Year)	Document Type	Location	Study Design	Diagnosis (Term Used)	Sample		Quality Score
					Age	N	
[1] ([1])	Art.	Colombia	Quant.	Borderline intelligence	Adoles.	114	4.95
[2] ([2])	Art.	Saudi Arabia	Quant.	Slow learners	Adoles.	120	4.89
[17] ([17])	Art.	Italy	Case	BIF	Child.	1	4.42
[16] ([16])	Art.	Italy	Quant.	BIF	Child./Adoles.	BIF = 106 MID = 168	4.79
[20] ([20])	Chapt.	Sweden	Theo.	BIF	No sample		3.84
[29] ([29])	Conf.	Malaysia	Lit.	Slow learners	No sample		4.00
[36] ([36])	Art.	Poland	Theo.	BIF	No sample		4.00
[41] ([41])	Art.	Indonesia	Case	BIF	Child.	1	4.89
[42] ([42])	Art.	Korea	Quant.	Borderline intelligence	Child.	SL = 15 CA = 18 LA = 19	4.68
[45] ([45])	Art.	Pakistan	Quant.	Slow learners	Child.	8	4.79
[53] ([53])	Art.	Indonesia	Mix.	Slow learners	Child.	61 (17 SL)	4.63
[67] ([67])	Art.	Malaysia	Theo.	Slow learners	No sample		4.53
[76] ([76])	Art.	India	Quant.	Slow learners	Child./Adoles.	20	4.95
[79] ([79])	Art.	Italy	Quant.	BIF	Child.	BIF = 85 TD = 45	4.74
[86] ([86])	Art.	Sweden	Quant.	BIF	Child.	BIF = 27 TD = 28	4.84

Note: Chapt. = book chapter; Conf. = conference paper; Art = research article; Child. = childhood; Adoles. = adolescence; Case = case study; Mix. = mixed methods; Quant. = quantitative; Lit. = literature review; Theo. = theoretical study; SL = slow learners; CA = chronological age-matched; LA = language age-matched; TD = typical development.

#### 3.2.2. Social Domain

Following the [9] ([9]) and the [6] ([6]), the social domain is composed of issues such as awareness of others’ thoughts, empathy, interpersonal communication skills, friendship abilities, social judgment and responsibility, gullibility and naïveté, self-esteem, social problem solving, and the ability to follow rules/obey laws and to avoid being victimized.

Twenty-four documents dealing with this set of topics were found. Among them, four subgroups emerged: the judicial world or the ability to follow rules/obey laws, life adversities, family interactions, and social adaptation.

**Ability to follow rules/obey laws:** In the judicial field, we first find three articles addressing the prevalence of individuals with BIF or ID in prisons. [24] ([24]) found that 55.6% of suspects in police custody in the Netherlands had BIF. In Spain, [26] ([26]) focused on adult defendants. A total of 64% of their sample had BIF (IQ 71–85). Lastly, [66] ([66]) analyzed the diagnoses of inmates in Austrian penitentiary institutions, detecting a prevalence of borderline intelligence of around 8%.

Additionally, the article by [10] ([10]) aimed to study the relationship between youth criminal histories, adult criminal histories, and sex offense characteristics, specifically focusing on incarcerated men with BIF or ID. They concluded that the BIF group was less likely to have a history of violence and to use violence in sexual offenses compared to the ID group. Meanwhile, [15] ([15]) interviewed community practitioners. These professionals noted that individuals with BIF faced challenges in maintaining positive social and family relationships and required some level of advocacy.

In the field of assessment tool development, the study published by [49] ([49]) explored the predictive validity of the Multiplex Empirically Guided Inventory of Ecological Aggregates for Assessing Sexually Abusive Risk for Coarse Abusive Children and Adolescents (MEGA♪). The cross-validation revealed that youth with BIF are at greater risk of engaging in inappropriate or abusive sexual behaviors compared to youth with typical development. However, they were significantly less likely to reoffend.

Furthermore, [13] ([13]) evaluated how a group of children with intellectual disabilities, BIF, and typical development fared during a mock cross-examination. Their findings show that nearly all children changed at least one response from their initial interview when challenged. No significant differences were found among children with ID, BIF, or TD regarding the likelihood of changing their responses.

Leaving behind the studies with population-based samples, [33] ([33]) conducted a review of legislative and policy documentation and concluded that, in the UK, the clinical definitions of ID tend to exclude individuals with BIF, which results in their exclusion from access to support services. Finally, a guide directed at state security bodies and forces was published by [22] ([22]). It aims to raise awareness, facilitate the detection of individuals with BIF, and provide guidelines for interacting with this population to ensure their fundamental rights and improve their quality of life.

Therefore, the ability to follow rules or laws appears to be a deficit in the population with BIF, with high prevalence rates of individuals in prison or engaging in criminal activities. For this reason, specialized attention in this sector is necessary.

**Life adversities:** Some articles address life adversities, that is, life conditions that negatively affect individuals and constitute risk factors for increasing their vulnerability. Examples include maltreatment, traumatic events, family dysfunction or neglect, or even belonging to a group with low socioeconomic status.

The studies by [14] ([14]) and [30] ([30]) are consistent in their findings, indicating that children with BIF experience more adversities than their peers with TD. For example, they are more likely to face social and/or material deprivation, adversities related to health events ([30]), educational problems, family disruption, and/or intervention by social services due to their parents’ inability to cope with the situation ([14]). Additionally, [90] ([90]) highlight that the prevalence of adverse childhood experiences is higher in children with BIF than in children with ID.

Moreover, [88] ([88]) propose a new perspective that highlights the role of self-confidence and attachment in understanding the special needs of children with BIF and ACE, as well as in tailoring rehabilitation interventions. They observed that there was a predominance of an attachment profile on the border between intermediate and insecure, characterized by low self-confidence and high separation anxiety, with a tendency towards somatization.

Additionally, [72] ([72]) observed that most of children with BIF required educational support in kindergarten or grade school, and this need for special pedagogical support increased with age. The risk of dropout seemed to increase during secondary education, which may indicate that the support provided at this educational stage is insufficient for students with BIF to graduate. Data beyond the educational stage were limited but showed an increased risk of unemployment and difficulties in maintaining a job.

In summary, individuals with BIF seem to be exposed to significant and varied life adversities, making them a particularly vulnerable population.

**Family interactions:** [19] ([19]) found that mothers of children with BIF and ID had significantly lower levels of education compared to mothers of children with TD. Additionally, by the age of 5, parents of children with BIF exhibited less positive engagement with their children and a higher rate of negative controlling behavior than parents of children with TD and even those with ID. Meanwhile, [35] ([35]) concluded that in the case of children with BIF, a dominant and rigid parenting style was significantly correlated with motivation for learning and locus of control for academic success. Interestingly, these correlations were not observed among children with typical development. On the other hand, [64] ([64]) studied that mothers of children with BIF reported significantly higher levels of overall parental stress, stress related to parenting, stress associated with a difficult child, and stress linked to parent–child interaction compared to mothers of control children.

Lastly, [50] ([50]) examined the implications of being a parent with BIF. Several areas of cognitive, reasoning, and emotional–behavioral skills were identified as potential areas of concern, including attention, memory, organization and time management, problem-solving, impulsivity, frustration/anger, and self-esteem.

Therefore, family interactions in the BIF population are characterized by complex relationships, seemingly rigid parenting styles, and high parental stress.

**Social adaptability:** Other articles investigated topics related to social adaptation. [25] ([25]) observed that the BIF group exhibited higher rates of poor social functioning and was more likely to have psychiatric diagnoses and to use drugs compared to a control group with average intelligence. In contrast, [93] ([93]) examined the effect of peer influence on prosocial behavior in adolescents with MID or BIF. Similar characteristics were reported across both. However, differences were identified in the effect of peer feedback on prosocial behavior, which was found to be greater in the BIF group than in the MID group.

Additionally, the issue of bullying was also addressed through a case study, where self-confidence and attachment were also investigated ([51]). The article showed the effectiveness of an intervention in reducing social anxiety symptoms and a bullying situation in a girl with BIF.

Finally, [94] ([94]) showed that individuals with BIF had significantly less social contact with friends, less strong confiding/emotional support from the closest person, and fewer confiding relationships with anyone compared to individuals with an IQ above 85, but more social contact with relatives. Also, it was observed that all social relationships were significantly related to a better quality of life for individuals with borderline intellectual functioning.

The social aspect is therefore another major area of difficulty for individuals with BIF, leading to additional risk situations such as psychiatric diagnoses, substance abuse, mental health problems, and challenges in quality of life.

**Theory of mind:** The study of by [11] ([11]) revealed that lower performance in theory of mind was exhibited by children with BIF compared to a control group. Moreover, these deficits in theory of mind were found to be closely connected to executive functions and meta-representational competences. Later on, [91] ([91]) found that primary school students with BIF exhibit a deficit in social understanding rather than in self-regulation. The authors suggest that this deficit in theory-of-mind skills is likely attributable to a borderline level of verbal intelligence.

**Table 2 ejihpe-15-00040-t002:** Main characteristics of documents addressing the social domain.

Author (Year)	Document Type	Location	Study Design	Diagnosis (Term Used)	Sample		Quality Score
					Age	N	
[10] ([10])	Art.	Australia	Quant.	Borderline ID	Adult.	ID = 31 BID = 13	4.79
[11] ([11])	Art.	Italy	Quant.	BIF	Child.	BIF = 28 TD = 31	4.58
[13] ([13])	Art.	UK	Qual.	Borderline ID	Child.	Moderate ID = 18 BID = 13 TD = 59	4.37
[14] ([14])	Art.	Italy	Quant.	BIF	Child.	BIF = 42 TD = 18	4.84
[15] ([15])	Art.	Australia	Qual.	BIF	Adult.	13	4.05
[19] ([19])	Art.	USA	Mix.	BIF	Child.	TD = 111 mothers/99 fathers BIF = 24/19 DD = 37/30	4.79
[22] ([22])	Gray	Spain	Pract.	Borderline intelligence	No sample		4.05
[24] ([24])	Art.	Netherlands	Quant.	Borderline IQ	Adult.	BIF = 99 ID = 51 TD = 28	4.84
[25] ([25])	Art.	Israel	Quant.	BIF	Adoles.	BIF = 76,962 Average IQ = 96,580	4.42
[26] ([26])	Art.	Spain	Quant.	BIF	Adult.	BIF = 74 ID = 26	4.74
[30] ([30])	Art.	UK	Quant.	BIF	All	ID = 426 BIF = 2108 Average IQ = 12,919	4.79
[33] ([33])	Art.	UK	Lit.	Borderline ID	No sample		4.37
[35] ([35])	Art.	Poland	Quant.	BIF	Child./Adoles.	BIF = 21 TD = 21	4.74
[49] ([49])	Art.	USA	Quant.	Low intellectual functioning	Child./Adoles.	BIF = 238 Not BIF = 818	4.79
[50] ([50])	Chapt.	UK	Theo.	Borderline LD	No sample		3.95
[51] ([51])	Art.	Spain	Case	Borderline intellectual ability	Child.	1	4.89
[64] ([64])	Art.	Italy	Case	BIF	Child.	BIF = 26 TD = 53	4.84
[66] ([66])	Conf. Abst.	Austria	Quant.	Borderline intelligence	Adult.	Not available	3.63
[72] ([72])	Art.	Finland	Quant.	BIF	All	651	4.89
[88] ([88])	Art.	Italy	Quant.	BIF	Child.	22	4.68
[90] ([90])	Art.	Netherlands	Quant.	BIF	Child./Adoles.	Severe ID = 6 Moderate ID = 20 Mild ID = 56 BIF = 52	4.84
[93] ([93])	Art.	Netherlands	Mix.	BIF	Adoles.	BIF = 19 MID = 21	4.79
[94] ([94])	Art.	UK	Quant.	BIF	All	IQ > 85 = 11,625 BIF = 2234 MID = 235	4.63

Note: Chapt. = book chapter; Conf. Abst. = conference paper abstract; Art = research article; Child. = childhood; Adoles. = adolescence; Adult. = adulthood; All = all ages; Case = case study; Mix. = mixed methods; Quant. = quantitative; Qual. = qualitative; Lit. = literature review; Theo. = theoretical study; Pract. = practical guide; BID = borderline intellectual disability; DD = developmental delay; TD = typical development.

#### 3.2.3. Practical Domain

The practical domain includes skills such as activities of daily living, personal care, job responsibilities or occupational skills, healthcare, money or telephone management, recreation, setting schedules or routines and organizing tasks, self-management of behavior, and use of transport ([9]; [6]). In this case, 12 documents were identified and categorized into employment, access to services, daily activities, quality of life, and risk factors.

**Employment**: In the field of employment, four documents were identified. [18] ([18]) investigated the association between employment and health in individuals with intellectual disabilities and borderline intellectual functioning. Individuals with both intellectual disabilities and BIF exhibited markedly lower employment rates and poorer health compared to other participants across all waves of data collection. In turn, [46] ([46]) found that employers in the industrial sector had limited knowledge about BIF. The main barrier identified was a lack of financial support, while the main advantages highlighted included corporate social responsibility and the company’s image and reputation. Regarding the work-related challenges faced by individuals with BIF, the study emphasized limited job training and inadequate task organization skills.

A book focused on the labor market inclusion of individuals with BIF was also found ([58]). The document defines borderline intellectual functioning (BIF), its characteristics, and labor rights. It offers guidance for employers on hiring and workplace adaptations, raises awareness among coworkers, explores recruitment processes, and shares personal work experiences of individuals with BIF.

Finally, a legal document related to the employment of people with BIF was included, specifically the Spanish Royal Decree 368/2021, of May 25, which outlines positive action measures to promote employment access for individuals with borderline intellectual capacity.

In summary, the employment area requires special attention in this population, which has low employment rates, engages in low-skilled jobs, and requires institutional support.

**Access to services:** [60] ([60]) found that individuals with BIF used significantly fewer services designed for people with disabilities than those with mild ID. In terms of disability pensions, individuals with BIF received more pensions than the general population, although fewer than those with mild ID. Additionally, individuals with BIF were hospitalized in psychiatric institutions more frequently and stayed longer than their typically developing peers. The authors concluded that individuals with BIF are at risk of unemployment and mental health issues and emphasized the need to raise societal awareness about BIF.

[97] ([97]) showed that, in Flanders, England, and Ontario, individuals with BIF either lack access to publicly funded ID care or find it more challenging to access these services than in the Netherlands, where individuals with borderline intellectual disabilities are classified as people with intellectual disabilities. However, access to these services is generally granted if additional conditions are present, such as autism, health issues, or homelessness.

Finally, [80] ([80]) presented a case study. Although the individual exhibited significant functional limitations compared to the general population, access to services was highly restricted due to the absence of a disability diagnosis. This case illustrates the importance and complexity of diagnosis in the BIF population, emphasizing that categorizing individuals solely based on their IQ fails to account for their daily adaptive abilities.

Thus, access to services emerges as a major barrier that complicates the lives of individuals with BIF due to their situation of being in a kind of no man’s land, between those with intellectual disabilities and those without.

**Quality of life:** In [81] ([81]), no direct relationship was identified between attending a special education school or mainstream education and success or quality of life in adulthood. The only variable investigated that predicted these outcomes in adulthood was the socioeconomic status of the families. Regarding parental-perceived health-related quality of life, [38] ([38]) observed that students with BIF exhibited a significant large deficit in the total score.

**Risk factors**: Some articles focused on other risk factors that influence people with BIF. [63] ([63]) found that the proportions of ID and BIF were significantly higher in the homeless population compared to the general population. On the other hand, [40] ([40]) concluded that BIF is a condition that generates significant costs, both in terms of indirect costs borne by parents—such as annual income loss or reduced productivity due to work absences to care for their children—and direct costs (medical and non-medical). Risk factors in girls consulting for early marriage were also studied ([54]). Nearly 20% of these girls were diagnosed with BIF, which was identified as one of the risk factors.

**Table 3 ejihpe-15-00040-t003:** Main characteristics of documents addressing the practical domain.

Author (Year)	Document Type	Location	Study Design	Diagnosis (Term Used)	Sample		Quality Score
					Age	N	
[18] ([18])	Art.	UK	Quant.	BIF	Young.	ID = 426 2,108 = BIF Higher IQ = 12,919	4.79
[40] ([40])	Art.	India	Quant.	BIF/Slow learners	Child./Adoles.	100	4.74
[38] ([38])	Art.	India	Quant.	BIF/Slow learners	Adult.	100	4.89
[46] ([46])	Art.	Spain	Qual.	BIF	NS	9 (companies)	4.89
[54] ([54])	Art.	Türkiye	Quant.	Borderline intelligence	Adoles.	80 ADHD = 15 Conduct D = 4 Anxiety D = 3 Depressive D = 2 Adjustment D = 1 Intellectual Disability = 6 BI = 16	4.89
[58] ([58])	Gray	Spain	Pract.	Borderline intellectual ability	No sample		3.68
[60] ([60])	Art.	Finland	Quant.	BIF	All	BIF = 416 MID = 312 LP = 284	4.89
[63] ([63])	Doct.	USA	Meta-a.	Borderline intelligence	Adult.	25 studies (N = 2.357)	4.79
[68] ([68])	Law	Spain	Law	Borderline intellectual ability	No sample		3
[80] ([80])	Conf.	Scotland	Case	BIF	Young.	1	3.74
[81] ([81])	Art.	Poland	Quant.	BIF	All	49	4.89
[97] ([97])	Gray	Netherlands	Lit.	Borderline ID	No sample		4.63

Note: Conf. = conference paper; Doct. = doctoral thesis; Art = research article; Gray = grey literature; Child. = childhood; Adoles. = adolescence; Young = emerging and young adulthood; Adult. = adulthood; All = all ages; Case = case study; Meta-a. = meta-analysis; Quant. = quantitative; Qual. = qualitative; Lit. = literature review; Pract. = practical guide; ADHD = Attention Deficit/Hyperactivity Disorder; BI = borderline intelligence.

#### 3.2.4. Miscellanea: Several Domains Addressed

Some documents addressed multiple aspects of adaptive functioning and were categorized in the scoping review by [56] ([56]) as belonging to several domains simultaneously. For example, [89] ([89]) analyzed the effects of a school-based emotion regulation intervention. While the intervention did not improve social skills or the parent–child relationship, it significantly enhanced academic performance. [23] ([23]) examined the relationship between BIF and borderline personality disorder (BPD). They found higher unemployment rates in individuals with both diagnoses compared to those with only BPD. Social adaptability was also lower in the comorbid group. Furthermore, [78] ([78]) conducted a literature review on the academic and socioemotional deficits of children with BIF. They noted widespread academic challenges, including difficulties in reading, writing, and mathematical reasoning, alongside working memory and attention deficits. Socioemotionally, these children faced school-based socialization challenges leading to isolation. This isolation contributed to adult problems, such as insecurity in the workplace, challenges in personal life, and potential psychopathologies.

[75] ([75]) highlighted that BIF population faces challenges in understanding social dynamics, stigmatization, and struggles with independent living, including household management, finances, and social navigation. Furthermore, society often rejects them, partly due to a lack of understanding, prejudice, and stigmatization. In the educational context, bullying stands out as one of the most painful experiences, which, together with academic underachievement, leads to a significant decrease in self-confidence and exclusion from the community. These challenges extended into adulthood, with high unemployment, underemployment, and workplace marginalization, thereby exacerbating socioeconomic inequalities and diminishing quality of life. The data in [61] ([61]) support the findings of [75] ([75]). They indicate that individuals with BIF exhibit lower rates of high school completion, fewer social relationships, and higher unemployment compared to the general population.

Additionally, six documents classified as “general” in the previously published scoping review ([56]) have also been selected for inclusion in this article. Although these papers discuss BIF in a broad manner, it seemed noteworthy to extract the specific information they provide regarding the adaptive functioning of this population. Therefore, as explained earlier, only the results related to the conceptual, social, and practical domains are presented below.

[21] ([21]) published an extensive report where they identified difficulties in the areas of language and communication. Academic failure was a widespread experience, accompanied by isolation and, in some cases, bullying. Most students with BIF were enrolled in mainstream education, typically in inclusive classrooms with additional support. In the social domain, family relationships were generally positive, but challenges included low persistence in the face of difficulties, heightened anxiety and anger responses, shyness and depressive tendencies. Furthermore, a high percentage demonstrated limited autonomy in leisure activities. In the practical domain, challenges included employability, independent travel, money management, and administrative tasks.

In the study of [34] ([34]), the educational environment was identified as a site of initial frustrations and often the first recognition of BIF. The report also identified a very high level of labor inactivity, with precarious access to the job market, inadequate job opportunities, and a business sector largely unaware of this condition. Regarding independent living, mobility was reported to be very low, with recognition of the need for external support to achieve independence. Inclusive leisure activities—typically facilitated by associations—emerged as the most common form of social participation. Risk behaviors, such as tolerating abuse, drug use, and pregnancy risks, were also observed. Finally, the report highlighted a pervasive lack of societal awareness and sensitivity toward BIF.

Meanwhile, [48] ([48]) argued that individuals with BIF exhibit poor academic performance, deficiencies in literacy and numeracy, low self-esteem, and difficulties forming emotional bonds. Additionally, these individuals face significant barriers to entering the labor market, and when they do, it is typically in low-skilled jobs. The issue of limited access to services resurfaced, as these individuals often lack formal disability recognition and, therefore, cannot benefit from social resources that could greatly assist them.

[27] ([27]) conducted an update on the concept of BIF and provided a critical analysis of the future of this diagnostic category, including aspects of adaptive functioning. Greenspan criticized the common practice of evaluating adaptive functioning using standardized instruments that often neglect descriptive and qualitative information. He recommended that the assessment of adaptive functioning should be a diagnostic requirement, complementing IQ measurements and focusing particularly on social judgment. Social judgment is notably impaired in this population and is a critical factor for key skills such as employment and independent living. The author also highlighted the high prevalence of BIF in contexts of criminality and poverty.

[31] ([31]) noted difficulties in executive functions, speed of information processing, working memory, declarative learning and memory, remote recall of information, temporal sequencing, and visuospatial functioning. In the social domain, issues included impulsivity, poor social judgment, and relationship struggles, which extended to practical difficulties like criminal behavior. The authors emphasized that individuals with BIF face gaps in services, as they are often ineligible for specialized intellectual disability services but also struggle to access generic services.

Also, the iconic CONFIL working group presented a consensus process for the “Consensus Manual on Borderline Intellectual Functioning” (not included in this systematic review due to its publication date but deeply considered in its elaboration) and its summary conclusions ([69]). The article highlighted low academic performance and difficulties in reading, writing, and mathematics, as well as attention deficits, low self-esteem, and lack of personal initiative. Furthermore, it stressed challenges in the labor market and the lack of social support services targeted at this population. The need to assess both the environment and adaptive capacities of individuals with BIF to conduct a comprehensive evaluation is emphasized.

Finally, only two articles that specifically addressed the construct of adaptive functioning were found. One of them, published by [37] ([37]), focused on the validation of the ADAPT (The ADaptive Ability Performance Test) in individuals with ID and BIF. The difference in ADAPT mean scores was significant across all IQ groups (moderate ID, mild ID, BIF, below-average IQ). Additionally, statistically significant differences were found in the total ADAPT score between the mild ID group and the BIF group, but not between the BIF group and the below-average IQ group.

On the other hand, [95] ([95]) compared adaptive functioning in a sample of youth referred for neuropsychological evaluation, comparing groups with average IQ, low average IQ (80–89), below-average IQ (70–79), and intellectual disability. The authors demonstrated that both the low and below-average IQ groups exhibited lower adaptive and academic functioning than individuals with average IQ. They recommend assessing adaptive functioning in clinical practice to ensure rigorous and accurate diagnoses and to develop effective support and interventions. Specifically, for individuals with an IQ of 70–89, demonstrating adaptive deficits could be a tool to help them access support services that they might not qualify for based solely on their IQ.

In summary, several documents encompass multiple areas of adaptive functioning simultaneously, such as studies that investigate different aspects at once or compare difficulties in the lives of individuals with BIF and other comorbid diagnoses. Additionally, there are publications with a more general perspective that address BIF in a holistic manner. These studies show how adaptive functioning, despite involving several areas, is a construct in which all its elements are inter-related and must be considered as a whole when studying and intervening in this population in an integrated way.

**Table 4 ejihpe-15-00040-t004:** Main characteristics of articles on adaptive functioning covering various domains.

Author (Year)	Document Type	Location	Study Design	Diagnosis (Term Used)	Sample		Quality Score
					Age	N	
[21] ([21])	Gray	Spain	Mix.	Borderline intelligence	All	BIF = 190 Families = 189	4.47
[23] ([23])	Conf.	Italy	Quant.	BIF	Adult.	BIF and BPD = 27 Only BIF = 25	3.63
[27] ([27])	Art.	USA	Theo.	BIF	No sample		4.21
[31] ([31])	Chapt.	UK	Theo.	BIF	No sample		4.32
[34] ([34])	Gray	Spain	Mix.	Borderline intellectual ability	All	11,619 Interviews: BIF = 3 Family = 3 Experts = 3	4.63
[37] ([37])	Art.	Netherlands	Quant.	BIF	Adult.	Moderate ID = 261 Mild ID = 617 BIF = 440 Below-average IQ = 60	4.89
[48] ([48])	Art.	Spain	Theo.	Borderline intellectual ability	No sample		4.31
[61] ([61])	Art.	Finland	Quant.	BIF	All	BIF = 156 MID = 170 LPs = 91	4.89
[69] ([69])	Art.	Spain.	Theo.	BIF	No sample		4.74
[75] ([75])	Art.	Spain	Mix.	BIF	Adoles.	30	4.95
[78] ([78])	Art.	Greece	Lit.	BIF	No sample		3.95
[89] ([89])	Mast.	Canada	Quant.	BIF	Child./Adoles.	24	4.84
[91] ([91])	Conf.	Russia	Quant.	BIF	Child.	BIF = 17 TD = 17	4.63
[95] ([95])	Art.	USA	Quant.	Low and below-average IQ	Child./Adoles.	Average IQ = 968 Low–average IQ = 639 Below-average IQ = 523 ID = 386	4.74

Note: Chapt. = book chapter; Conf. = conference paper; Gray = gray literature; Mast. = Master’s thesis; Art = research article; Child. = childhood; Adoles. = adolescence; Adult. = adulthood; All = all ages; Mix. = mixed methods; Quant. = quantitative; Lit. = literature review; Theo. = theoretical study; BPD = borderline personality disorder; LPs = learning problems; TD = typical development.

### 3.3. Limitations Reported by the Included Studies

The limitations most frequently pointed out by the papers included in this review relate to the population samples, which were relatively small in most cases and, in some instances, likely biased. This inevitably limits the generalizability of the results. Additionally, weaknesses in the methodological design of the studies are evident, including the lack of control groups and randomization, as well as the lack of longitudinal studies.

Some studies also highlight limitations in the assessment tools used, pointing out incomplete validation, potential evaluator bias, or floor effects, as well as the constructs measured, suggesting that additional variables could have been evaluated to make the studies more comprehensive and in-depth.

The lack of available scientific evidence on topics related to BIF is also noteworthy, making it challenging to build new studies in this field. Finally, some articles identify the failure to directly assess adaptive functioning as a limitation. This is a crucial aspect for diagnosing intellectual disabilities and BIF, and its omission may lead to less precise sample selection.

## 4. Discussion

This systematic review has facilitated the study of the available knowledge in both the scientific and gray literature regarding the adaptive skills of individuals with BIF. Most of the studies employed quantitative methodologies and focused on the child population. Notably, only three studies, all conducted in Spain and two of them by third-sector organizations, directly engaged individuals with BIF to explore their lives, analyzing their discourse qualitatively and delving into their opinions and experiences ([21]; [75]; [34]). On the other hand, several studies adopting a quantitative perspective report limitations such as small sample sizes and constraints in measuring variables. Moreover, there is significant terminological variability in this field. While the term “borderline intellectual functioning” is the most used, numerous other terms are also employed to refer to this condition.

As previously stated, the aim of this article is to develop a profile of adaptive skills in the BIF population based on the existing literature. However, it is evident that not all articles address the intrinsic characteristics of individuals with BIF, which would constitute the profile itself. Some also discuss external factors, more related to societal structures, such as care services, interventions, and legislation. Therefore, this body of knowledge can be divided into two categories: first, the profile of adaptive skills specific to individuals with BIF; and second, the societal barriers and resources that affect individuals with BIF, limiting their adaptive functioning.

### 4.1. Adaptive Functioning Profile in the BIF Population

From the analysis of the articles included in this systematic review, there emerge certain characteristics that, collectively, could outline the profile of the BIF population in terms of adaptive skills. In general, individuals with BIF exhibit lower adaptive skills compared to the general population ([95]) but higher than those of individuals with mild intellectual disability ([37]).

More specifically, in the **conceptual domain**, individuals with BIF exhibit widespread academic difficulties, including challenges in reading, writing, metaphorical reasoning, mathematics, short- and long-term memory, working memory, and attention ([16]; [20]; [21]; [31]; [36]; [42]; [48]; [69]; [78]; [79]; [86]). These difficulties, coupled with a lack of attention, support, and awareness within the educational system, often lead to low academic performance or academic underachievement ([1]; [21]; [31]; [75]; [86]; [95]). Additionally, individuals with BIF are at high risk of school dropout ([61]; [72]) and experience significant frustration, low self-esteem, and behavioral problems ([34]; [48]; [69]).

These findings are supported by high-level scientific studies not included in this review due to their publication date being prior to the inclusion period ([70]; [39]). Likewise, the systematic review by [59] ([59]) presents consistent results in this area, indicating that individuals with BIF show deficits in memory, working memory, reading, and arithmetic skills, which result in low academic performance upon completing compulsory education. Considering this information, early identification in schools could be possible by disseminating this knowledge among teachers and counselors. Additionally, the implementation of educational methodologies and support for students with BIF would be feasible, thereby reducing school failure, low self-esteem, and frustration.

The **social domain** also surfaces as a key area of difficulty for individuals with BIF. This population demonstrates poor social functioning and struggles to understand social and family dynamics ([23]; [25]; [27]; [31]; [34]; [75]; [93]). These difficulties may be a consequence of low theory-of-mind skills ([11]; [91]). As a result, individuals with BIF tend to maintain limited social contact and relationships and receive insufficient social and emotional support ([15]; [31]; [34]; [48]; [61]; [94]). Their autonomy in leisure activities is also limited ([21]), potentially due to low personal initiative ([69]). Moreover, individuals with BIF tend to experience greater life adversities or adverse childhood experiences compared to the general population ([14]; [30]; [88]), and in some cases, even more than individuals with intellectual disabilities ([90]). These challenges contribute to high levels of parental stress among their caregivers ([64]) and dominant, controlling parenting styles characterized by less positive engagement ([19]; [35]). This lack of social understanding, combined with barriers to accessing services and the absence of interventions, has been linked to a high prevalence of criminal behavior among individuals with BIF ([24]; [26]; [27]; [31]; [66]). Additionally, they are more likely than the general population to engage in risky behaviors, such as inappropriate or abusive sexual activities ([49]), although this is less frequent than among individuals with intellectual disabilities ([10]). Other risk factors include tolerance of abuse, drug use, and pregnancy risks ([34]).

The findings from [59] ([59]) also align with these results. According to this review, children with BIF engaged in more solitary play and showed poorer social information processing compared to their typically developing peers. Regarding parenting, [59] ([59]) observed that mothers of children with BIF exhibited less positive engagement and less sensitive parenting than mothers of children with average intelligence. Similarly, [70] ([70]) emphasize this area, highlighting that the difficulties individuals with BIF face in establishing social relationships are caused both by their own challenges and by the rejection attitudes of the wider society. Having a group of friends, hobbies, and fulfilled emotional connections are essential aspects for leading a meaningful and fulfilling life.

In the **practical domain**, employment and independent living skills appear to be areas of significant difficulty. Individuals with BIF have low employment rates, often working in low-skilled and precarious jobs ([18]; [21]; [23]; [27]; [34]; [48]; [60]; [61]; [69]; [75]), and struggle with organizational skills and the training required for employment ([21]; [46]). Similarly, they face significant challenges in living independently, particularly in the areas of household management and financial skills ([21]; [27]; [75]). Specific situations for which BIF is considered a risk factor include homelessness ([63]) and undesired marriages ([54]). Lastly, encompassing the aspects, the quality of life and health-related quality of life of individuals with BIF may be compromised ([38]; [81]).

In contrast, findings from [59] ([59]) regarding the practical domain differ from those reported here. While they suggest that employment rates among individuals with BIF were like those of typically developing individuals, albeit with lower wages and less prestigious and lower-skilled jobs, the current systematic review clearly indicates lower employment rates among the BIF population. However, [59] ([59]) caution that the reliability of their results in this area is limited due to the outdated nature of the studies included and the small sample sizes. Additionally, while they discuss marriage rates, they do not frame BIF as a risk factor for undesired marriages. Instead, they report that individuals with BIF marry or cohabitate at slightly lower rates than those with average intelligence. Meanwhile, [70] ([70]) support the argument that access to the labor market is particularly challenging for individuals with BIF. Additionally, they frequently highlight the difficulties associated with independent living and the heightened risk of abuse or manipulation.

In Figure 2, we can observe the adaptive skills profile in individuals with BIF in a graphical format, highlighting the main characteristics within each of the conceptual, social, and practical domains.

### 4.2. How Does Society Treat Individuals with BIF? Barriers and Resources

In addition to the characteristics inherent to the population with BIF, arising from their cognitive, behavioral, and adaptive profile, interaction with society plays a key role in their daily lives. Therefore, the barriers and resources available within society will have a fundamental mediating effect, either promoting or decreasing the autonomy and well-being of this population. This systematic review has identified various social barriers and resources that directly impact individuals with BIF.

#### 4.2.1. Barriers

The main barrier identified through the review of the documents included in this systematic review is the institutional exclusion that individuals with BIF face in accessing support and care services ([31]; [33]; [48]; [60]; [69]; [75]; [80]; [81]; [97]). Access requirements for these services are determined by the possession of a disability certificate, and the reliance on IQ-centric definitions of disability leads many individuals with BIF to be deemed ineligible for this certificate. Therefore, despite facing numerous limitations and difficulties in their daily lives, these individuals find it very difficult to access services or benefit from assistance. Additionally, the presence of an individual with BIF exponentially increases the family’s economic costs, both direct and indirect, spent on aspects such as healthcare, school support, or re-education therapies ([40]), partly due to the difficulty of accessing public resources.

A noteworthy case in the sphere of access to services is the Netherlands, where BIF is included under the intellectual disability classification, thus guaranteeing access to services ([55]; [93]; [96]; [97]). This could be an approach that could be extrapolated to other countries to facilitate the support that individuals with BIF so desperately need.

Furthermore, there are two major opposing viewpoints regarding the future classification of BIF. While both agree on the need to bridge the gap between high prevalence and low recognition, some argue that BIF should be considered a substantial disorder as it has specific characteristics that require specialized attention ([96]). While, on the other hand, some advocate for raising the threshold of Criterion A (intellectual functioning) for diagnosing intellectual disability from 70 to 85 ([27]). In this case, a person would be eligible for an ID diagnosis if their IQ is below 85 and, in addition, they present deficits in adaptive functioning.

However, it is necessary to reconsider the weight that IQ holds in current diagnoses. Although the concept of IQ no longer appears in diagnostic manuals, in practice and even in research, it continues to be a widely used number. Authors such as [12] ([12]) argue that continuing to define conditions based solely on IQ levels is problematic because intellectual disability disorders are multifactorial, and skills can vary between individuals with the same IQ score. Therefore, they suggest that more attention should be paid to performance in more specific cognitive functions. A thorough evaluation of adaptive functioning is, once again, key in these processes, as it would allow for a much more precise diagnosis of needs and, therefore, the efficient allocation of the necessary resources.

In addition to experiencing institutional exclusion, individuals with BIF sometimes suffer from social rejection, often due to the limited general knowledge about this condition ([34]; [46]; [75]). In schools, bullying emerges as one of the most painful experiences for children with BIF ([51]; [75]). This, along with their difficulties in comprehension and social relationships, exacerbates the cycle of social isolation.

#### 4.2.2. Resources

However, society’s treatment of individuals with borderline intellectual functioning is not entirely negative. There are guides or manuals written by both academic experts and third-sector organizations that aim to raise awareness about BIF in various fields such as employment ([58]), law enforcement ([22]), and even parenting ([50]). Additionally, there are legislative documents starting to emerge that attempt to regulate this situation, such as the Spanish Royal Decree 368/2021, which outlines measures to promote employment for this population.

Furthermore, several studies have shown effectiveness in improving various skills, such as reading, mathematics, compliance, task completion, developmental skills, and academic functioning or outcomes. These studies utilized different methodologies, including computer programs, academic interventions, music therapies, or school-based emotion regulation interventions ([2]; [17]; [41]; [45]; [76]; [89]). In addition, other theoretical proposals have been developed to take BIF students into account for the implementation of measures such as the use of tablet technology or digital game-based applications ([29]; [67]), and alternative methods to the psychometric approach have been proposed to facilitate the early identification of BIF students ([53]).

Finally, associations stand out as significant resources where individuals with BIF and their families can find support, solutions, engage in leisure activities, and socially interact ([21]; [34]).

## 5. Conclusions

This systematic review has enabled the construction of a profile of adaptive skills in individuals with borderline intellectual functioning (BIF) based on the literature (see Figure 2). Additionally, it has highlighted future research directions, including the need to conduct studies with larger, more representative samples and to delve deeper into the experiences and perspectives of individuals with BIF.

A potential limitation of this study lies in the possibility of having overlooked relevant documents addressing this topic. Despite employing an extensive and meticulous search strategy, the lack of terminological consensus regarding this construct complicates research efforts. Therefore, it is essential for the scientific community to reach an agreement on what constitutes BIF and how it should be defined and named.

Nevertheless, the findings clearly demonstrate that individuals with BIF face challenges across all three domains of adaptive functioning. Yet very few studies explicitly reference this construct in their research. In fact, some researchers cite the absence of adaptive functioning assessments as a limitation of their studies ([79]; [93]). This oversight may stem from two main factors: first, the lack of specific criteria for BIF in official diagnostic manuals, which diminishes the perceived need to assess both intellectual and adaptive functioning in this population; and second, the scarcity of standardized tools calibrated for individuals with BIF. Indeed, most tools used to assess adaptive functioning are primarily designed for individuals with intellectual disabilities and rely on normative samples that include small groups of adults without disabilities. As a result, the ceiling (i.e., the highest level of adaptive functioning that these tests can measure) is often adjusted only for individuals with ID, impacting the categorization of abilities in the broader population ([73]).

Considering this information, early identification in schools could be achievable by disseminating these insights among teachers and school counselors, thereby enabling the implementation of educational methodologies and support systems tailored to students with BIF. Such measures could reduce school failure, low self-esteem, and frustration. Similarly, employment and social support services must be made aware of the daily challenges faced by individuals with BIF to provide them with appropriate resources aimed at enhancing their autonomy and improving their quality of life.

Lastly, this systematic review could serve as a foundation for shaping legislation to recognize BIF. Although legislative efforts have begun to emerge, it is imperative to address the gap that individuals with BIF face in accessing services, ensuring that they no longer remain in a state of limbo.

## Figures and Tables

**Figure 1 ejihpe-15-00040-f001:**
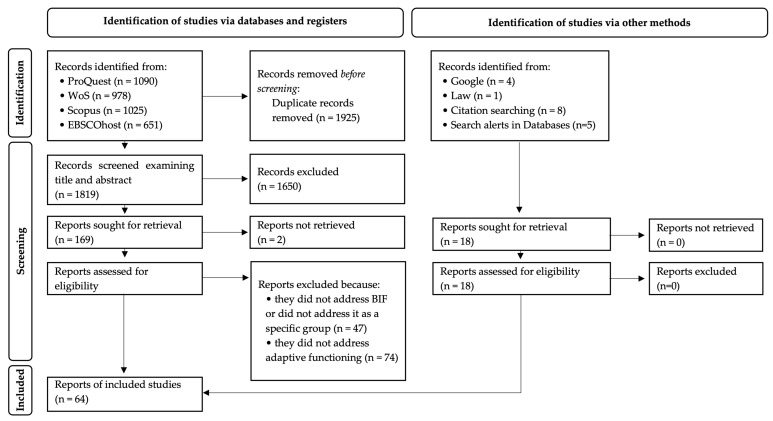
Flow diagram of study selection process. From: [57] ([57]).

**Figure 2 ejihpe-15-00040-f002:**
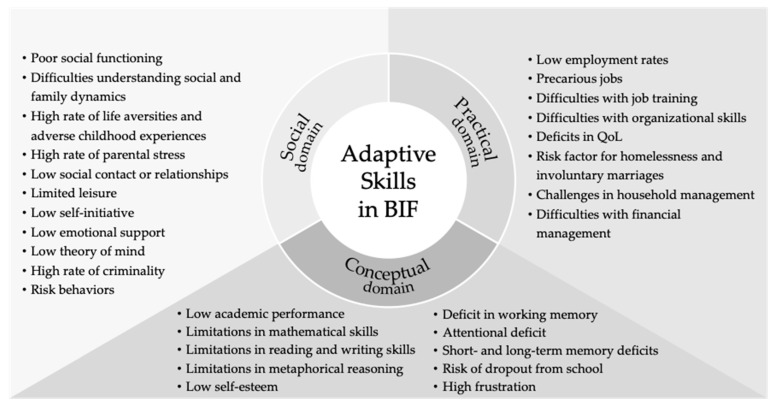
Adaptive Skills Profile in BIF.

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
