# Peer review of "Understanding Adaptive Skills in Borderline Intellectual Functioning: A Systematic Review"

_ejihpe, 2025, doi:10.3390/ejihpe15030040_

Round 1
Reviewer 1 Report
Comments and Suggestions for Authors
This manuscript is a systematic review of adaptive functioning in individuals with borderline intellectual functioning (BIF). Articles were included from 2012 onwards that studied BIF as a specific group. 64 documents were reviewed and included in the current review. The article summarizes adaptive functioning by each domain (i.e., conceptual, practical, and social)
This manuscript addresses a complex topic that has not been clearly addressed in the literature. The Methods are clearly communicated, and the authors were thoughtful in how they presented the results of very different studies. The authors were able to consider the practical implications for this often over-looked group of individuals. Despite these strengths, I have some concerns about several aspects of the manuscript, as detailed below.
- Beginning on line 46, the authors give a brief summary of another review article aimed at understanding BIF. The summary of that article is unclear and because the article is under review, readers cannot look it up. The disconnect is when the authors say that “these documents were categorized […].” Can the authors clarify this paragraph?
- The authors provide a very detailed historical account of adaptive functioning in the introduction. It may be more effective to condense this information. The paragraph starting on line 70 and the one on line 78 could be integrated into one shorter paragraph.
- In the Methods section (line 137), the authors state, “The scoping review included a total of 138 documents published between January and June 2024 regarding borderline intellectual functioning.” Are they talking about the current review or the previous one? Also, I am not understanding how 138 documents were published between January and June 2024. Can the authors please clarify?
- The paragraph starting on line 152 helps to clarify some of my previous comments, but this issue should still be addressed earlier in the manuscript.
- In the section 3.2.1, the section on “Theoretical proposals” is not well integrated and is unclear as to how it relates to characterizing adaptive functioning.
- At the conclusion of the “Academic profiles” section beginning on line 297, please include a summary paragraph to help readers understand the main take-aways from the studies reviewed.
- If the authors want to keep the “Interventions” section, they should include a stronger rationale for how it relates to other overall purpose of the article in the introduction. I’m not sure it really adds much to the review.
- Authors should define “life adversities” before reviewing first study – line 509
- How the authors determined the bolded headers in each adaptive functioning domain is not entirely clear. Please add further information in the Methods section.
- Broadly, I think the manuscript could be much improved if the Results section was substantially shortened and better integrated. This section lists and reviews studies without helping readers to synthesize the information.
- The Discussion section better integrates the data, so I wonder if the authors would want to organize the results section more succinctly into tables. Tables would make the article easier to read and not as redundant from the results to discussion sections.
Author Response
Thank you very much for your comments, which we believe have significantly improved the article. Please refer to the attached file to view our responses.

Reviewer 2 Report
Comments and Suggestions for Authors
The contribution deals with an interesting and underinvestigated topic, such as the adaptive behavior in individual with a border intellectual functioning (BIF).
The Author (s) summarized and discussed recent evidence from studies according to the conceptual, social and practical domains, in which adaptive functioning is articulated, showing that they present difficulties in all areas.
Although the review is well conducted, it is somehow not clearly focused, to the extent that it assumes the skills that constitute the prerequisites of adaptive behavior correspond to adaptive behavior itself. The querry string used confirm this biased assumption.
In fact the correct definition of adaptive behavior construct is centered on the observable and measurable performances and not on underlying skills involved: “the performances of daily activity required for personal and social sufficiency […. ] Finally, adaptive behavior is defined by typical performance, not ability. While ability is necessary for the performance of daily activities, an individual adaptive behavior is inadequate if the ability is not demonstrated when it is required” (Sparrow, Cicchetti & Balla, 2003, pag. 6).
The Author (s) correctly recognized in the discussion that no evidence is available of the measurement of adaptive behavior in individual with BIF, due to the lack of specific, standardized psychometric tools.
To avoid confusion or erroneous expectations in the reader, it would be appropriate to specify the construct of the adaptive behavior in the introduction, underlying its distintive characteristics and clarifying that the review refers to studies on skills that support adaptive behavior and not on actual adaptive behavior, without research evidence.
Therefore I suggest, first of all, to modify the contribution title to better specify the topic addressed, such as “adaptive functioning skills” and not simply “adaptive functioning”. and, consistently to use, throughout the entire text, adaptive skills in place of adaptive functioning.
Minor revisions:
Please check the following sum (lines 262-266): n. 48 (population-based samples)= 15 + 6+ 8+2+9+7?
Author Response

(The authors gave the same response as above.)

Reviewer 3 Report
Comments and Suggestions for Authors
The article is a broad meta-analysis of studies on borderline intellectual functioning (BIF). It highlights interesting aspects of the topic studied, particularly by highlighting the authors' imperfect alignment regarding using the term BIF. Furthermore, the analysis shows a greater sensitivity to the topic in some geographical areas compared to others. The authors, after introducing the theoretical and practical aspects that revolve around the concept of BIF, illustrate in great detail the selection process of the articles subjected to investigation, making the reader aware of the rigor with which the selection was conducted. In terms of analysis of the results, they identify 3 crucial domains for people affected by BIF and illustrate, through the analysis of quantitative and qualitative studies, the main problems that individuals with BIF encounter in daily life, underlining how in almost all the countries to which the analyzed works refer, subjects with BIF are poorly protected, finding themselves in a gray area between "normal" intellectual development and intellectual deficiency.
The meta-analysis is conducted with rigor and, from a methodological point of view, there are no particular observations to make. The only evident flaw of this work is the objective difficulty of reading with particular reference to the part relating to the results. The text is composed of micro sentences followed by abundant and redundant bibliographical references that do not encourage reading. A greater "lightness" of the writing style would certainly favor the diffusion of the article.
Author Response
First of all, thank you very much for reviewing this manuscript, for the considerable time spent on it, and for sharing your knowledge regarding BIF and adaptive functioning.
Following your suggestions as well as those of the other reviewers, the entire results section has been significantly summarized and rewritten to integrate the information and enhance its clarity. Additionally, short paragraphs highlighting the main findings have been included at the end of each bolded header. We hope this synthesis facilitates reading and improves the understanding of the article. Please let us know if further improvements are needed.
Round 2
Reviewer 1 Report
Comments and Suggestions for Authors
Thank you for taking the time for revisions. I believe you addressed my major concerns, and only a few minor edits can be made.
Line 175-176 - I suggest rewording this sentence and no using the word "reorganize" and the readers do not know what you are "reorganizing." "Organize should suffice. Additionally, it remains unclear about what "determined inductively" means. How did you arrive at these topic headers? Are they common themes in the literature? Where they explicitly mentioned in several of the articles? The last part of the sentence in line 176-77: "...with the aim of..." is not needed.
Author Response
Comments 1: Line 175-176 - I suggest rewording this sentence and no using the word "reorganize" and the readers do not know what you are "reorganizing." "Organize should suffice. Additionally, it remains unclear about what "determined inductively" means. How did you arrive at these topic headers? Are they common themes in the literature? Where they explicitly mentioned in several of the articles? The last part of the sentence in line 176-77: "...with the aim of..." is not needed.
Response 1: Thank you once again for taking the time to review this article. The sentence in lines 174-177 has been rewritten following your suggestions. It is highlighted in red in the attached manuscript.
Thank you once again. Thanks to your suggestions and those of the other reviewers, we believe the article has significantly improved.
Reviewer 2 Report
Comments and Suggestions for Authors
The revision of original contribution has been well improved. Therefore this current version is suitable for publication
Author Response
Thank you once again for taking the time to review this article. Thanks to your suggestions and those of the other reviewers, we believe the article has significantly improved.